# Osteocytes: New Kids on the Block for Cancer in Bone Therapy

**DOI:** 10.3390/cancers15092645

**Published:** 2023-05-07

**Authors:** Aric Anloague, Jesus Delgado-Calle

**Affiliations:** 1Department of Physiology and Cell Biology, University of Arkansas for Medical Sciences, Little Rock, AR 72205, USA; aanloague@uams.edu; 2Department of Physiology and Cell Biology, Winthrop P. Rockefeller Cancer Institute, University of Arkansas for Medical Sciences, Little Rock, AR 72205, USA

**Keywords:** osteocytes, cancer, bone, sclerostin, myeloma, breast cancer, metastasis, therapy

## Abstract

**Simple Summary:**

This review describes how osteocytes, the most abundant cells in bone, act in cancers that grow in bone. It summarizes how cancer cells reprogram osteocytes to participate in processes associated with tumor growth, cancer cell survival, migration, angiogenesis, and bone destruction. These changes ultimately facilitate the spread of cancer in the bones and promote bone disease. It also discusses some of the identified signaling pathways mediating these processes and the therapeutic strategies to target them.

**Abstract:**

The tumor microenvironment plays a central role in the onset and progression of cancer in the bone. Cancer cells, either from tumors originating in the bone or from metastatic cancer cells from other body systems, are located in specialized niches where they interact with different cells of the bone marrow. These interactions transform the bone into an ideal niche for cancer cell migration, proliferation, and survival and cause an imbalance in bone homeostasis that severely affects the integrity of the skeleton. During the last decade, preclinical studies have identified new cellular mechanisms responsible for the dependency between cancer cells and bone cells. In this review, we focus on osteocytes, long-lived cells residing in the mineral matrix that have recently been identified as key players in the spread of cancer in bone. We highlight the most recent discoveries on how osteocytes support tumor growth and promote bone disease. Additionally, we discuss how the reciprocal crosstalk between osteocytes and cancer cells provides the opportunity to develop new therapeutic strategies to treat cancer in the bone.

## 1. Introduction

Bone is a highly dynamic tissue continuously resorbed by osteoclasts and formed by osteoblasts to maintain tissue homeostasis and integrity. This process, known as “bone remodeling,” requires the coordinated and balanced action of bone resorption and formation. Osteocytes, long-lived descendants of fully differentiated osteoblasts, orchestrate bone remodeling [1,2,3]. During the activation phase of bone remodeling, osteocytes detect microdamage and initiate local bone resorption [1,3,4]. Although the cellular and molecular mechanisms are not fully deciphered, osteocyte apoptosis contributes to a cascade of events that result in the formation of a canopy that encloses the bone remodeling compartment [2,5]. Osteocyte apoptosis can also locally increase the expression of pro-osteoclastogenic genes, such as RANKL, M-CSF, or HMGB1, generating a microenvironment that favors the recruitment and differentiation of osteoclasts [2,4,6,7,8]. Once osteoclasts mature, the bone resorption phase occurs. In this process, osteoclasts create a resorption pit as they secrete hydrochloric acid, cathepsins, tartrate-resistant acid phosphatase, and matrix metalloproteinases (MMPs). These enzymes work together to break down the crystalline hydroxyapatite and organic matrix compounds [4,9]. Following bone resorption is the reversal phase of bone remodeling, in which resorption switches to formation. In this phase, osteoclasts supply coupling factors that recruit osteoblast precursors to the resorption pit. During this phase, Wnt signaling is activated, stimulating osteoblastogenesis and mature osteoblasts’ bone-forming function [10]. In addition to their ability to initiate local bone resorption, osteocytes also regulate bone formation through the secretion of sclerostin and DKK1-1, potent Wnt signaling antagonists and inhibitors of osteoblast differentiation/function [10]. Mature osteoblasts synthesize and secrete osteoid, a matrix enriched in type 1 collagen [9]. Finally, during the termination phase of bone remodeling, osteoblasts either undergo apoptosis, become lining cells covering inactive bone surfaces, or differentiate into osteocytes [9].

The bone provides fertile soil and a complex biological system for cancer cells to thrive. Cancers that start in the bone are known as primary bone cancers. Primary bone cancers are uncommon, occur from cells present in bone tissue, and include osteosarcomas, chondrosarcomas, Ewing sarcomas, and chordomas [11]. Other cancers, including multiple myeloma or leukemias, can start in the bone but do not originate from bone cells (i.e., multiple myeloma originates from plasma cells in the bone marrow). Finally, metastatic bone tumors can develop from cancer cells that metastasize to the bone from other organs, such as breast or prostate cancer. Depending on the type and stage of cancer, patients with tumors in the bone can present with increased fracture risk, spinal cord compression leading to neuropathies, mineral dysregulation, and severe bone pain. These factors significantly contribute to morbidity and mortality in affected individuals [12,13,14].

The growth of cancer cells in bone profoundly affects the bone marrow microenvironment and disrupts the balance of bone remodeling. The thriving of cancer cells in bone has been traditionally explained by the establishment of a “vicious cycle” between tumor cells and cells of the bone microenvironment that fuels tumor growth and bone disease. In lytic cancers (i.e., multiple myeloma or breast cancer), bone colonizing tumor cells secrete factors that promote osteoclastogenesis (i.e., PTHRP, IL6, RANKL, and M-CSF), tilting the balance in bone remodeling towards bone resorption [15,16,17,18]. As a result, bone lytic lesions are formed, and the integrity of the bone is compromised. The rapid bone resorption induces the release of growth factors from the mineral matrix (i.e., activin, TGFβ, FGF, and PDGF), promoting tumor growth. Cancers such as osteosarcoma or metastatic prostate cancer create osteoblastic lesions. These tumor cells stimulate osteoblast differentiation, leading to the rapid and exuberant formation of bone tissue with compromised mechanical properties [11,19,20]. In turn, osteoblasts produce growth factors that promote cancer cell proliferation [19,20].

Given the central role of cells of the bone marrow in the onset and progression of cancer in the bone, research efforts have focused on finding aberrant cellular and molecular mechanisms in this niche. This led to the identification of new cellular interactions between cancer cells and cells of the bone marrow (i.e., adipocytes, immune cells, stromal cells, etc.) beyond those originally described in the “vicious cycle” paradigm. Mounting evidence supports that osteocytes, overlooked in cancer for decades, generate a microenvironment conducive to tumor progression and bone disease (Figure 1). Unlike osteoblasts, osteoclasts, or other bone marrow cells that have relatively shorter lifespans, osteocytes have a remarkably long half-life. This unique characteristic makes them not only abundant (constituting 95% of cells in bone) but also a durable source of signals in the bone-tumor microenvironment. Although embedded in the mineral matrix, osteocytes have long cytoplasmic processes that emanate from the cell body and run through the canaliculi arising from the lacunae, forming a syncytial inter-communication network known as the osteocyte lacuno–canalicular system [21]. Through this system, osteocytes establish contact and engage in communication with cells present at the bone surfaces and with neighboring osteocytes. Further, this system enables cell communication through gap junctions between the cytoplasmic processes and extracellular communication through the fluid that passes through the lacuno–canalicular system [21]. Osteocytes utilize this network to distribute soluble factors and establish physical interactions with other cells, regulating the behavior of distant cells. In this review, we discuss the functions of osteocytes in bone cancer. Additionally, we highlight new opportunities for targeting osteocytes and their derived factors to control tumor progression and improve bone health in cancer patients.

## 2. Role of Osteocytes on Epithelial–Mesenchymal Transition, Migration, and Invasion of Cancer Cells

Breast and prostate cancer metastases to bones are relatively common and a leading cause of cancer deaths in the U.S. To achieve metastasis to a secondary site, cancer cells undergo a multi-step process that begins with detachment from the primary tumor, followed by migration through the extracellular matrix of the surrounding tissue to enter the bloodstream (invasion–intravasation–dissemination) [22]. To metastasize, tumor cells leave the bloodstream, migrate through the extracellular matrix, and invade the bone (extravasation–colonization–homing) [22]. In the initial steps of this process, metastatic cancer cells undergo epithelial-to-mesenchymal transition (EMT). This transformative process involves the loss of their epithelial characteristics and the acquisition of a more mesenchymal phenotype, resulting in morphological changes and increased cellular plasticity [23]. As discussed below, the role of osteocytes in the early stages of metastasis to bone is unclear.

### 2.1. Prostate Cancer

Osteocytes produce GDF15, which supports prostate cancer migration and invasion [24]. Osteocytes also express different matrix metalloproteinases (i.e., MMP2, MMP13, and MMP14), which could assist metastatic cancer cells in migrating through the extracellular matrix and invading the bone. Osteocytes, as mechanosensitive cells, are stimulated by various mechanical forces within the bone. They are capable of translating mechanical signals into biological cues to coordinate local bone formation and bone resorption. As a consequence of that, osteocytes play a vital role in maintaining the integrity of the skeleton and adapting it to meet the mechanical demands placed upon it [25]. Several studies, mostly using in vitro systems, have investigated the impact of osteocyte mechanical loading on EMT, migration, and invasion of cancer cells. In prostate cancer cells, applying hydrostatic pressure to osteocytes stimulated their migration [26]. In contrast to this study, conditioned media from mechanically stimulated osteocytes decreased the expression of EMT-related genes and the migration/invasion capacity of prostate cancer cells [26,27,28].

### 2.2. Breast Cancer

In vitro studies show that conditioned medium from osteocytes induces EMT and increases the migration of breast cancer cells [29,30]. Mechanistically, these effects appear to be mediated by changes in CX43, SNAIL, and WISP signaling, as well as adenosine nucleotides released by osteocytes [31,32,33,34,35]. Osteocyte-derived osteopontin has also been associated with the activation of EMT in breast cancer cells. Further, conditioned media from osteocytes treated with bisphosphonates reduced breast cancer migration, an effect dependent on osteocyte-derived ATP [31]. However, the ATP bi-product, adenosine, promoted the growth of MDA-MB-231 cells, suggesting distinct roles for ATP and adenosine released by osteocytes [31]. Further, co-injection of osteocytes with Wnt signaling activation and breast cancer cells inhibited brain tumor metastasis [36], which is another common site of metastasis in breast cancer. The impact of mechanically stimulated osteocytes on the migration and EMT of breast cancer cells has also been studied, with conflicting results. In one study, factors released by mechanically stimulated osteocytes enhanced the migration of cancer cells [37,38]. Using a cytokine array, osteocyte-derived CXCL1 and CXCL2 cytokines were identified as regulators of migrating breast cancer cells [38]. Contrary to these findings, conditioned media from mechanically stimulated osteocytes decreased the expression of EMT-related genes and the migration/invasion capacity of breast cancer cells [26,27,28]. Likewise, conditioned media from osteocytes exposed to oscillatory fluid flow reduced endothelial permeability and adhesion of breast cancer cells by decreasing MMP9 expression [39]. These effects were dependent on the intensity of the mechanical stimulation [27]. Lastly, osteocytes exposed to low-intensity vibration, a mechanical signal that mimics the dynamics of muscle contraction, suppressed the migration and invasion of breast cancer cells [40,41]. 

The different outcomes observed in studies investigating the effects of mechanically stimulated osteocytes on prostate/breast cancer cells are probably due to the utilization of different mechanical stimuli (such as stretching, pressure, and vibration) and cell culture models that do not fully reproduce the complex mechanosensory role of osteocytes in the bone. 

### 2.3. Multiple Myeloma

Migration and invasion are also important for hematological cancers [42]. However, the mechanisms regulating cancer cell movement within the bone marrow and metastasis to secondary sites in the hematological system are largely unknown. Using an in vivo intratibial model, Trotter et al. found that osteocyte ablation using diphtheria toxin promoted the migration and homing of myeloma cells from one tibia to the bone marrow of the contralateral limb [43]. This suggests that osteocytes can influence the migration of myeloma cells. It is important to note that changes in osteoblast and osteoclast populations, bone loss, and increases in bone marrow adiposity accompany the systemic ablation of osteocytes induced by diphtheria toxin. Thus, it is difficult to identify the cellular mechanism(s) responsible for this observation in this mouse model.

Collectively, these results suggest that osteocyte-derived factors could influence the metastatic potential of cancer cells at primary sites and generate a microenvironment favorable for recruiting and retaining cancer cells in bone. However, the existing data supporting this concept is primarily derived from in vitro studies, which do not fully represent the complexity of inter-organ communication. Moreover, the in vivo evidence supporting this notion remains limited. Nonetheless, given the emerging data showing that osteocytes are endocrine cells and have regulatory functions beyond the skeleton (kidney, muscle, nervous system, peripheral fat) [2], we cannot entirely exclude the participation of osteocytes in the early steps of metastasis. Further research on this topic is needed to clarify the role of osteocytes in the onset of secondary metastasis to bone.

## 3. Role of Osteocytes in Tumor Growth

Accumulating evidence supports the idea that osteocytes have an active role in tumor growth in metastatic and hematological cancers. Osteocytes affect tumor growth via different mechanisms, including the production of growth factors, physical interactions with cancer cells, and fueling the resorption that drives the “vicious cycle” of bone cancer. Yet, the effects of osteocytes on cancer growth seem to be cancer-specific. 

### 3.1. Prostate Cancer

Conditioned media from MLO-Y4 osteocytes cultured in 2D or 3D conditions stimulates the proliferation of human prostate cancer cell lines [29]. This suggests the existence of a soluble factor capable of promoting cancer cell proliferation. Osteocytes produce GDF15 [24], and the knockdown of this gene in bone diminished the proliferation of prostate cancer cells injected into the tibiae of mice. In another study, applying pressure to osteocytes induced prostate cancer growth in an in vivo mouse model, partly by upregulating CCL5 and matrix metalloproteinases in osteocytes [44]. In addition to soluble factors, osteocytes can influence tumor growth via physical interactions. Using high-resolution imaging and histological approaches, different groups have shown that cell-to-cell communication between cancer cells and osteocytes in the bone niche occurs. Confocal imaging identified direct contact between prostate cancer tumor cells in the bone marrow and osteocytes [45]. However, the impact of these physical interactions between osteocytes and prostate cancer cells on tumor growth has not been explored. 

### 3.2. Breast Cancer

As described for prostate cancer cells, breast cancer cells can also establish cell-to-cell contact with osteocytes [45]. However, in vitro studies have shown that cell-to-cell co-culture with MLO-Y4 osteocyte-like cells did not increase the proliferation of breast cancer cells [46]. The role of osteocyte-derived factors in breast cancer tumor growth is ambiguous (recently reviewed by Pin et al. [11]). For instance, in vitro studies show that conditioned media from MLO-A5 osteocyte-like cells increased the proliferation of breast cancer cell spheroids, whereas cell-to-cell culture conditions decreased it [27,29,34]. Culture of MLO-A5 cells in gelatin hydrogels, overexpression of LRP5, or mechanical stimulation turned the osteocytes into suppressors of breast cancer cell growth [27,33,36,47,48]. Contrary to these studies, conditioned media from mechanically stimulated MLO-Y4 osteocyte-like cells increased breast cancer cell growth by producing CXCL1 and CXCL2 [38]. Lastly, osteocytes protected breast cancer cells from the proliferative inhibitory effects of oxidative stress, which are dependent on CX43 signaling [46]. 

The disparate outcomes of these studies are likely a consequence of different culture models (2D vs. 3D), different osteocyte-like and cancer cell lines, and different culture conditions. Investigation of the role of osteocytes in in vivo/ex vivo models containing authentic osteocytes in the tumor niche might better elucidate the effects of these cells on tumor proliferation and the migration and invasion potential of bone metastatic cancer cells. 

### 3.3. Multiple Myeloma

Contact between osteocyte dendritic processes and myeloma cancer cells has been reported using acid etching and electron microscopy [49]. These physical interactions are partially mediated by Notch signaling. The Notch signaling pathway is activated by cell-to-cell contact between adjacent cells and regulates proliferation, apoptosis, and differentiation in multiple cells of the bone/bone marrow [50]. Notch signaling is dysregulated in several cancers due to alterations in the expression of Notch receptors or ligands [51], shaping how cancer cells interact with their surroundings. In multiple myeloma, reciprocal Notch signaling exists between osteocytes and myeloma cells. Notch activation by osteocytes increases CYCLIN D1 expression and enhances the proliferation of myeloma cells [49]. Mechanistic studies revealed that osteocyte proliferative signals are mediated by NOTCH3 in myeloma cells. Genetic knockdown of NOTCH3 in myeloma cells prevented the increase in CYCLIN D1 expression and proliferation induced by osteocytes [52]. Unlike the studies described for prostate or breast cancer, the impact of osteocyte-derived soluble factors on myeloma growth remains unknown. 

The findings that osteocytes can impact cancer cell proliferation in bone have sparked interest in targeting these cells and their derived factors to stop tumor growth.

### 3.4. Mechanical Stimulation of Osteocytes

Consistent with the evidence that mechanical signals alter osteocytes’ ability to influence cancer cell proliferation, several studies have shown that mechanical stimulation can decrease tumor growth in bone in metastatic and hematological cancers [27,33,53,54,55,56]. The exact mechanisms underlying these effects are not fully understood, including whether they are direct actions of the mechanical signals on cancer cells, indirect actions mediated by osteocytes or other mechanoresponsive cells, or a combination of both.

### 3.5. Inhibition of Osteocyte-Derived Notch Signals

The crosstalk between osteocytes and myeloma cells via Notch has also proven to be an attractive therapeutic target to stop tumor growth. In an in vivo model of multiple myeloma, inhibition of NOTCH3 signaling in myeloma cells, which is the receptor mediating osteocyte–myeloma communication, resulted in a reduction in tumor size [52]. Similarly, pharmacologic inhibition of Notch signaling downstream of all the Notch receptors using a bone-targeted gamma-secretase inhibitor prevented osteocyte pro-proliferative effects on myeloma cells in vitro and in vivo [49,57]. 

### 3.6. Inhibition of Osteocyte-Derived Bone Remodeling Factors

Breaking the vicious cycle between tumor cell proliferation and bone resorption is another approach to mitigating tumor growth. As discussed in more detail in Section 5 and Section 6, targeting osteocyte-derived factors regulating bone resorption (i.e., RANKL) or formation (i.e., sclerostin or DKK-1) also has an impact on the progression of cancers that grow in bone.

## 4. Role of Osteocytes in Neo-Angiogenesis in the Bone Tumor Microenvironment

Bones are highly vascularized through a complex system of heterogeneous blood vessels (reviewed in [58,59]). Endothelial cells form an intricate network of blood vessels that sustain various microenvironments in the bone marrow to maintain mesenchymal and hematopoietic stem cells. The vasculature of the bone is essential for oxygen and nutrient supply and also influences bone remodeling. Angiogenesis, the process of new blood vessel formation from existing vasculature, is closely coupled with osteogenesis [60] and plays a crucial role in tumor survival and progression [61]. 

Cancer patients exhibit increased vascular density in the bone marrow niche, and marrow angiogenesis negatively correlates with patient survival [62,63,64]. Once colonized by cancer cells, the bone marrow microenvironment becomes hypoxic, facilitating angiogenesis and providing essential pro-angiogenic factors such as VEGFA [65]. Osteocytes can sense oxygen levels and induce vascular changes in the bone marrow [66,67]. In a recent study, the number of VEGFA^+^ osteocytes doubled in bones bearing murine and human myeloma tumors compared to saline-injected controls [68]. The percentage of VEGFA^+^ osteocytes positively correlated with Endomucin^+^, a marker of endothelial cells, and with tumor vessel area [68]. Consistent with these results, in vitro studies conducted on osteocytes cultured under hypoxic conditions mimicking their environment (1% O_2_) showed that myeloma cells upregulated the expression of several angiogenic factors, such as VEGFA, and enhanced the osteocytes’ ability to promote vessel formation [68]. The mechanisms behind VEGFA upregulation by myeloma cells in osteocytes are not fully understood. In vitro studies showed that myeloma cells increase FGF23 in osteocytes, which in an autocrine manner upregulates osteocytic VEGFA production [68]. Myeloma cells also activate Notch signaling in osteocytes via cell-to-cell interactions [49,57]. Notch signaling is a well-established regulator of angiogenesis and thus could also mediate the pro-angiogenic potential of osteocytes [69]. Together, these findings provide one of the first pieces of evidence supporting the idea that cancer cells induce a pro-angiogenic phenotype in osteocytes. Nonetheless, further studies are needed to determine the specific contribution of osteocyte-derived angiogenic factors to tumor angiogenesis in myeloma and other cancers that grow in bone. 

## 5. Role of Osteocytes in Cancer-Induced Osteoclastogenesis and Bone Resorption

One of the hallmark features of lytic cancers is the development of bone lesions, which lead to increased fracture risk, pain, disability, and morbidity [70,71]. Complex cellular and molecular mechanisms mediate the bone loss accompanying the growth of cancer cells in bone (reviewed in [16,72,73,74,75,76]). Most of the current insights regarding the role of osteocytes in cancer-induced bone destruction come primarily from myeloma studies. These data support the idea that cancer cells reprogram osteocytes to generate a microenvironment that favors bone resorption over bone formation. 

### 5.1. Prostate and Breast Cancer

One mechanism by which cancer cells stimulate resorption through osteocytes involves the induction of apoptosis and the alteration of their canalicular network [45,49,77,78]. These effects have been reported in prostate and breast cancer mouse models and have also been associated with age-related bone loss [3]. However, whether patients with breast/prostate metastases exhibit a higher prevalence of apoptotic osteocytes or the contribution of osteocyte apoptosis to the skeletal effects of secondary metastasis to bone has not been carefully investigated. 

### 5.2. Multiple Myeloma

Myeloma patients exhibit increased apoptotic osteocytes in areas infiltrated with myeloma cells [78,79]. This clinical observation has been recapitulated in mouse models of myeloma and in in vitro cell cultures [45,49]. Mechanistically, myeloma cells induce osteocyte apoptosis via TNFα and Notch signaling [49]. Apoptosis increases osteocytes’ osteoclastogenic potential by stimulating RANKL and IL-11 production and enhancing their ability to recruit pre-osteoclasts and form mature osteoclasts [49,78]. Inhibition of apoptosis partially prevented RANKL upregulation in osteocytes by myeloma cells in vitro [49], suggesting other mechanisms are involved. In this regard, myeloma cells produce 2-deoxy-D-ribose (2DDR), which activates the major histocompatibility complex class II transactivator (CIITA) in osteocytes [80]. CIITA activation acetylates histone 3 lysine 14 in the promoter of RANKL, leading to RANKL expression and osteoclastogenesis [80]. However, similar to inhibiting osteocyte apoptosis, blockade of CIITA signaling only partially prevented myeloma-induced RANKL upregulation in osteocytes [80].

Osteocytes also support bone destruction by stimulating RANKL production in cancer cells. For instance, osteocytes upregulate RANKL expression in myeloma cells via Notch signaling. Inhibition of NOTCH3 in myeloma cells reduced osteocyte-induced RANKL upregulation [52,57]. Consistent with this, injection of NOTCH3 knockdown myeloma cells into mice resulted in fewer lytic lesions [52,57]. However, in this study, NOTCH3 knockdown in myeloma cells suppressed tumor burden. Therefore, the authors could not determine whether the decreased resorption was due to RANKL downregulation, decreased tumor burden, or a combination of both. Thus, the debate about the major source of RANKL in the tumor microenvironment—osteocytes vs. cancer cells or other cells of the bone marrow (i.e., immune cells, adipocytes, osteoblasts, progenitors)—remains unresolved. 

### 5.3. Other Cancers

Another mechanism of osteocyte-driven bone destruction was recently identified in invasive oral squamous cell cancer (OSCC), which shows a bone-destructive phenotype, particularly in the maxillary of the mandibular bone. HMGB1 is an “alarmin” released by osteocytes that undergo apoptosis, and it has been shown to have a role in regulating osteoclastogenesis [81]. HMGB1 is highly expressed in OSCC cancers and increases osteoclastogenesis and bone resorption via induction of RANKL in osteocytes [81]. Interestingly, osteocytes also promote resorption in non-metastatic bone cancers, including colon adenocarcinoma, ovarian cancer, or Lewis lung carcinoma [82]. As in cancers that grow in bone, osteocyte apoptosis and dysregulation of the dendritic network were observed, accompanied by upregulation of several genes associated with bone resorption (ACP5, CTSK, ATP6V0D2, and MMP13) in osteocytes [82,83].

With the advancements in medical treatment for many cancers, patients are experiencing longer survival rates. As a result, effective management of bone disease has become paramount to reducing fractures and bone pain and improving the quality of life of cancer patients. Bisphosphonates and anti-RANKL are the mainstay therapeutic options to treat cancer bone loss [84,85]. Both therapies effectively decrease skeletal-related events and improve the quality of life of cancer patients. However, the sustained inhibition of bone resorption seen with bisphosphonates is associated with side effects (i.e., osteonecrosis of the jaw) [86]. Similarly, anti-RANKL discontinuation leads to a rebound phenomenon that increases resorption and fracture risk [87]. As discussed below, targeting osteocytes in the bone niche can become an alternative approach to stopping bone destruction in cancer of the bone. 

### 5.4. Mechanical Stimulation of Osteocytes

Because mechanical signals decrease the osteocyte’s potential to induce osteoclast formation [25], the osteocyte’s mechanosensitivity can also be exploited to interfere with cancer-induced bone destruction. For instance, the ability of myeloma cells to increase RANKL expression in osteocytes and osteoclastogenesis is mitigated by fluid shear stress in vitro [88]. In another in vitro study, low-intensity vibration or pharmacological stimulation of mechanical pathways using YODA1 reduced osteocyte apoptosis induced by breast cancer cells [89]. In an in vivo setting, mechanical stimulation applied to the tibia was found to preserve trabecular bone mass in a model of breast cancer with bone metastasis [90]. Moreover, Pagnotti et al. have shown in different models of bone cancer the beneficial effects of low-intensity vibration as a means to deliver mechanical signals to patients with compromised mobility. Low-intensity vibration reduced osteoclast surface and protected bone mass in mouse models of multiple myeloma and spontaneous granulosa cell ovarian cancer [40,53,91]. The same group has recently reported the protective effects of low-intensity vibration against the bone loss induced by complete estrogen deprivation therapy, frequently used in breast cancer patients [92]. As mentioned above, whether osteocytes mediate these effects is unclear and demands further investigation.

### 5.5. Inhibition of Osteocyte-Derived Notch Signals

In mouse models of multiple myeloma, pharmacologic inhibition of Notch signaling with a novel bone-targeted gamma-secretase inhibitor (BT-GSI) decreased RANKL expression and had potent anti-resorptive effects, similar to those seen with bisphosphonates [57]. Remarkably, BT-GSI did not interfere with physiological bone formation or reduce tumor growth [93]. Myeloma cells increase NOTCH3 in osteocytes [49], and genetic activation of NOTCH3 in osteocytes leads to bone loss [94]. Pharmacologic inhibition of NOTCH3 in vivo is sufficient to decrease bone resorption in naïve mice [95]. Therefore, the identification of NOTCH3 signaling as a mediator of RANKL and bone resorption in the tumor microenvironment suggests the potential use of anti-NOTCH3 antibodies as a therapeutic approach to halting bone destruction in the cancer setting. 

### 5.6. Inhibition of Osteocyte-Derived Bone Remodeling Factors

Using a combination of genetic approaches, different groups showed that osteocytes are a major source of RANKL in the adult bone under physiological conditions [4,6,7,96]. RANKL production by osteocytes is increased in cancer that grows in bone [49]. Thus, osteocytes, through RANKL, might promote the local bone resorption that fuels the initial stages of the vicious cycle of bone cancer. To date, whether osteocyte-derived RANKL (vs. RANKL from other cells) plays a major direct or indirect role in tumor growth has not been examined. Denosumab not only prevents skeletal-related events in cancer patients but has been associated with anti-tumor effects in solid and hematological malignancies [97,98]. To date, whether osteocyte-derived RANKL plays a major direct or indirect role in tumor growth has not been examined. An alternative approach to addressing bone resorption is through the pharmacological blockade of the osteocyte-derived Wnt antagonist sclerostin. This approach promotes bone formation and transiently reduces osteoclasts and bone resorption [99]. Consistent with this, the pharmacological inhibition of sclerostin is accompanied by suppression of bone resorption in some mouse models of multiple myeloma and metastatic breast cancer, although the effect may vary across different models [100,101,102,103,104]. The mechanisms underlying the decreases in bone resorption seen with anti-sclerostin are unclear, although there is some evidence that sclerostin can regulate RANKL expression [10,105].

## 6. Role of Osteocytes in Cancer-Induced Modulation of Bone Formation

Several mechanisms have been described for the dysregulation of bone formation in bones infiltrated with cancer cells (reviewed in [16,72,73,74,75,76]). Osteocytes and their derived factors have recently emerged as regulators of bone formation in the tumor microenvironment. High expression of Wnt antagonists (i.e., sclerostin and DKK-1) has been detected in cancer patients with bone involvement and mouse models of cancer in bone [49,100,102,106,107,108], suggesting osteocyte production of these factors is affected by cancer cells in bone. 

### 6.1. Prostate and Breast Cancer

It is important to note that whereas osteocytes can be the main source of Wnt antagonists in some cancers, in others, tumor cells also produce these factors [101]. This is the case with prostate and breast cancer cells. In prostate cancer, which typically forms osteoblastic lesions characterized by high bone formation, the levels of sclerostin, or DKK-1, can vary during disease progression [109], adding complexity to our understanding of the role of osteocytes in this process. In in vitro models, conditioned media from prostate cancer cells decreases the expression of sclerostin and DKK-1 in osteocytes [110,111]. These results suggest that the decrease in the production of Wnt antagonists by osteocytes creates a permissive niche for the formation of osteoblastic lesions. Sclerostin overexpression, but not DKK-1 overexpression, in prostate cancer cells decreased metastasis and migration of prostate cells [112]. Similarly, in breast cancer cells, genetic overexpression of sclerostin or DKK-1 leads to more bone metastases and bone disease [102,113]. Intriguingly, a recent study in a mouse model of breast cancer bone metastasis, which can lead to a mix of lytic and blastic lesions, showed that sclerostin production by cortical osteocytes is suppressed only in areas adjacent to tumor cells [103]. However, whether the decreased sclerostin production colocalizes with the presence of blastic lesions was not explored. Further studies are needed to determine the specific contribution of osteocyte-derived vs. tumor-derived Wnt antagonists to the dysregulation of bone formation in metastatic cancers. 

### 6.2. Multiple Myeloma

Serum sclerostin levels are increased in hematological cancers such as multiple myeloma [49,100]. Unlike breast or prostate cancer cells, myeloma cells do not express sclerostin [104]. In vitro, co-culture with myeloma cells increases SOST/sclerostin in osteocyte cell lines and authentic osteocytes. Conditioned media from osteocytes exposed to myeloma cells but not from control osteocytes decreased the expression of osteoblastic genes (RUNX2, ALPL, and COL1A) and osteoblast differentiation in vitro [49,108]. This suggests that osteocyte-derived sclerostin contributes to the profound suppression of osteoblasts by myeloma cells. The mechanisms regulating sclerostin production in multiple myeloma-bearing bones remain unclear. As discussed in Section 5, 2DDR production by myeloma cells and the consequent activation of CIITA in osteocytes are also partially responsible for the elevated SOST, the gene encoding sclerostin, expression in bones bearing myeloma tumors [80]. Another potential mechanism for the upregulation of sclerostin is DKK-1 signaling. DKK-1 is a Wnt antagonist overproduced by osteocytes and myeloma cells [80,114,115]. In vitro, pharmacological inhibition of DKK-1 prevented sclerostin upregulation by myeloma cells, and treatment with recombinant DKK-1 increased it [108]. To date, this hypothesis has not been tested in vivo. 

### 6.3. Other Cancers

The information available about the role of osteocytes in the bone formation accompanying osteoblastic lesions in osteosarcoma is very limited and demands further investigation. In primary bone cancers, the levels of sclerostin, or DKK-1, also vary with disease progression [109]. As described above for prostate cancer, sclerostin has anti-tumor effects in osteosarcoma, as administration of recombinant sclerostin inhibited tumor growth and migration and improved survival in a mouse model of osteosarcoma [116]. However, the effects of recombinant sclerostin on bone mass or microarchitecture were not studied.

Anti-cancer therapies have only minor effects, if any, on repairing damaged bones [117]. Thus, bone repair remains a major challenge in cancer management. Targeting Wnt signaling in common skeletal disorders has become an effective approach to overcoming bone loss [99]. However, because Wnt signaling is one of the key cascades regulating cancer onset and progression [118], there is hesitation about using Wnt signaling activators in the oncology setting. Below, we present some of the different anabolic stimuli tested in preclinical models of bone cancer and discuss their impact on tumor growth.

### 6.4. Mechanical Stimulation of Osteocytes

Mechanical signals are anabolic to bones and promote bone formation in areas of strain. Mechanical stimulation has been used successfully to boost osteoblast function in various mouse cancer models and cancer patients (recently reviewed in [119]). Because cancer patients exhibit compromised bone microarchitecture, their ability to practice physical activity can be challenging. In this regard, whole-body vibration devices that exert safe, low-intensity strain forces are an alternative for loading the skeleton in cancer patients [53,91,92,119,120]. 

### 6.5. Inhibition of Osteocyte-Derived Bone Remodeling Factors

With the development of neutralizing antibodies against DKK-1 [121], a new opportunity arose to repair damaged bone in lytic cancers. Preclinical studies in mouse myeloma models showed that targeting DKK-1 with neutralizing anti-DKK-1 antibodies increased osteoblasts, reduced multinucleated osteoclasts, and increased bone mass [122,123]. The bone anabolic effects of anti-DKK-1 may also be associated with reduced myeloma burden [122,123]. Despite the promising results in preclinical mouse models, the outcomes of the clinical trials were less robust [124], and anti-DKK-1 antibodies have not been FDA-approved for treating cancer patients. Of note, DKK-1 inhibition increases SOST/sclerostin expression in bone, which can counteract the anabolic effects of DKK-1 suppression [125]. Genetic or pharmacologic [126] deletion of both DKK-1 and SOST/sclerostin leads to a robust anabolic response greater than when inhibiting each gene alone [125]. Similarly, the co-administration of DKK-1 antibodies and a novel anti-LRP6 antibody also led to superior skeletal outcomes in mouse models of multiple myeloma [127]. To date, the effects of dual inhibition of sclerostin and DKK-1 in cancer models have not been investigated. 

Anti-sclerostin therapy was approved by the FDA in 2019 to treat postmenopausal women at high risk of fracture. The bone gain induced by the anti-sclerostin antibody results from opposing effects on bone formation (increased) and bone resorption (decreased) [128,129]. Preclinical animal studies showed that anti-sclerostin antibodies induce new bone formation in bones infiltrated with myeloma or breast cancer cells, increase bone mass, and improve bone mechanical properties [100,101,102,103,104,106,108]. Similar observations have been reported in immunodeficient SOST KO mice [100,103]. In certain mouse models, the blockade of sclerostin signaling has been shown to result in reductions in osteoclast activity, although this effect is not observed in all models [100,104]. To date, the cellular/molecular mechanisms by which anti-sclerostin releases the suppression of osteoblasts in the tumor microenvironment remain unclear. 

In multiple myeloma models, activation of Wnt signaling via pharmacological inhibition of sclerostin one day after cancer cell injection or after 3 weeks did not affect tumor growth when tumors were established [100,104,108]. Similarly, the injection of myeloma cells in SOST KO mice did not affect myeloma progression [100]. Lastly, Wnt activation via co-administration of a novel anti-LRP6 antibody and anti-DKK-1 did not affect tumor activity in a mouse model of myeloma disease [127]. Remarkably, anti-sclerostin did not alter the anti-tumor effects of chemotherapy [108], suggesting that it could be used as an adjuvant therapy to improve bone health in patients with multiple myeloma. All these studies have turned sclerostin into an emerging target for treating myeloma-induced lytic bone disease and prompted the investigation of sclerostin antibodies in a clinical trial with multiple myeloma patients (NCT05775094). 

In breast cancer bone metastasis, treatment with anti-sclerostin in a mouse model with established bone metastasis after cardiac inoculation reduced the growth of MDA-MB-231 breast cancer cells [101]. Moreover, anti-sclerostin treatment in mice with intrafemoral breast cancer tumors did not affect tumor growth rate but improved overall survival [106]. In addition, genetic deletion of SOST in breast cancer cells or administration of S6, a new inhibitor of sclerostin-STAT3 signaling, immediately following cardiac inoculation reduced the ratio of bone metastases [102]. However, in a recent study, administering anti-sclerostin one week before tumor inoculation resulted in no major changes in tumor volume/area [103]. Intriguingly, the authors of this manuscript reported that pre-treatment with anti-sclerostin increased tumor area only when MDA-MB-231 breast cancer cells were used [103]. The reasons for the discrepancy between the outcomes of pre-treatment (one week before cell inoculation) and treatment (once metastases were established) in mouse models injected with MDA-MB-231 breast cancer cells are unclear. However, this discrepancy raises the possibility that inhibition of sclerostin could transform the bone niche into a more favorable metastatic site. Nonetheless, this hypothesis contrasts with the observation that breast cancer metastases exhibited the same growth rate in control and SOST KO mice [103]. 

Given the remarkable beneficial skeletal effects observed in models of cancer-induced bone loss with therapies targeting Wnt antagonists, further studies, including clinical trials, are warranted. These studies aim to determine which cancer patients could benefit from these therapies, identify the ideal therapeutic window, and explore the most effective combinations that provide beneficial bone effects without increasing tumor growth or causing significant toxicity. 

## 7. Conclusions

During the last decade, we have witnessed major advances in our understanding of the role of osteocytes in bone cancer. Initially viewed as passive bystanders embedded in the mineral matrix, osteocytes are now considered key components of the tumor microenvironment. Nowadays, we have a better understanding that osteocytes play a significant role in influencing tumor growth, bone resorption, and bone formation in most cancers that develop in the bone. However, it is important to note that their impact, whether positive or negative, on each of these processes appears to vary depending on the cancer type. The knowledge acquired in this field has provided the rationale for developing new approaches to treat cancer in bone-targeting osteocytes and their derived factors (Figure 2). However, the effects of these therapies need to be carefully assessed for different cancer types. In conclusion, osteocytes contribute to the pathophysiological regulation of bone metabolism in cancers that develop in the bone. The identification of signals and pathways through which cancer cells alter osteocyte biology is vital to improving current therapeutic approaches and discovering novel strategies to target osteocytes. This pursuit aims to achieve superior and safer treatments for cancers that affect the bone.

## 8. Future Directions

Despite the tremendous advances in our knowledge of osteocyte biology in cancer, critical areas remain underexplored. For instance, a complete understanding of how osteocyte gene expression is affected by different cancers or the underlying regulatory mechanisms remains elusive. Unbiased genome-wide approaches, such as single-cell RNA sequencing (scRNAseq), have the potential to significantly advance our understanding of the molecular changes underlying osteocyte dysfunction. These approaches can aid in generating a gene signature that distinguishes healthy from diseased osteocytes, identifying new biomarkers with predictive value for bone disease, and uncovering novel therapeutic targets. Although scRNAseq enables the identification of differential gene expression within diverse cell populations, it does not provide information about cellular localization. Thus, techniques that provide spatial resolution (spatial transcriptomics or in situ hybridization RNAscope) could help us understand the interactions between osteocytes and cancer cells in a complex environment such as the bone marrow. Furthermore, there is a need for the development of reliable and clinically relevant mouse models (syngeneic) of cancer in the bone that facilitate genetic manipulation of the osteocytes, eliminating the need for extensive backcrossing across generations. Such models would provide valuable in vivo evidence to better understand the influence of osteocytes on the devastating effects of bone tumors.

Another area where the available data is limited is the impact of anti-cancer therapy on osteocyte function and its consequences. For instance, bortezomib, a proteasome inhibitor frequently used to treat myeloma, prevented osteocyte apoptosis in myeloma cells [79]. In contrast, Aplidin, a marine-derived compound with potent anti-myeloma efficacy, induced osteocyte apoptosis [130,131]. Similarly, radiation, typically used to treat metastatic cancers, induced the accumulation of senescent and apoptotic osteocytes and bone loss [132,133,134].

Once in bone, most invading cancer cells undergo apoptosis, with a few becoming dormant and surviving [135]. The cellular and molecular mechanisms regulating cancer cell dormancy in bone remain largely elusive; however, interactions between cancer cells and cells in the bone marrow and the vascular niche appear to be involved [136,137]. Osteoblasts and osteoclasts can awaken dormant cells in bone [137,138]. The role of osteocytes, which regulate osteoblasts and osteoclasts, in bone homing, dormancy, or early survival processes of cancer cells in the skeleton is unknown and demands further investigation. 

Lastly, the role of osteocytes in the onset of primary bone tumors, such as osteosarcoma, remains unexplored. Expression of osteocyte markers (i.e., DMP1, MEPE, or PHEX) has been observed in the osteoblastic subtype of human osteosarcoma, indicating that osteocytes may potentially serve as progenitors for osteosarcoma cells [139,140]. Supporting this notion, the injection of MLO-Y4 osteocyte-like cells in mice formed solid tumors with bone lesions similar to those observed in osteosarcoma patients [139]. Contrary to this hypothesis, Fgfr3^+^ endosteal stromal cells deficient in p53 can give rise to osteosarcoma-like lesions through unregulated self-renewal and aberrant osteogenic processes [141].

## Figures and Tables

**Figure 1 cancers-15-02645-f001:**
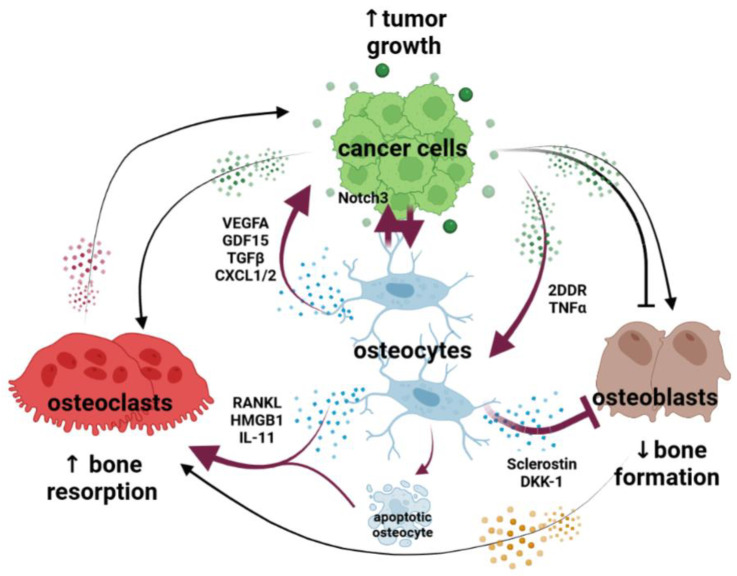
Osteocytes are at the center of the vicious cycle of osteolytic cancers. The growth of cancer cells in bone reprograms osteocytes, which become an abundant and durable source of signals in the tumor niche. Osteocytes communicate with tumor cells via physical interactions (Notch signaling, NOTCH3) and soluble factors (GDF15, TGFβ, CXCl1/2, VEGFA) that promote cell proliferation and the formation of a blood vessel network that supports tumor progression. Osteocytes undergo apoptosis via Notch activation and TNF signaling. Through factors secreted by cancer cells (2DDR), osteocytes overproduce pro-osteoclastogenic factors (i.e., RANKL, HMGB1, and IL-11) that further stimulate osteoclastogenesis, bone resorption, and bone destruction. Osteocytes in bones infiltrated with cancer cells secrete Wnt signaling antagonists (sclerostin and DKK-1), reducing osteoblast number and their bone-forming activity. Black lines denote the previously described “vicious cycle” interactions between osteoblasts, osteoclasts, and cancer cells. Magenta lines denote new cellular interactions between osteocytes, cancer cells, osteoblasts, and osteoclasts.

**Figure 2 cancers-15-02645-f002:**
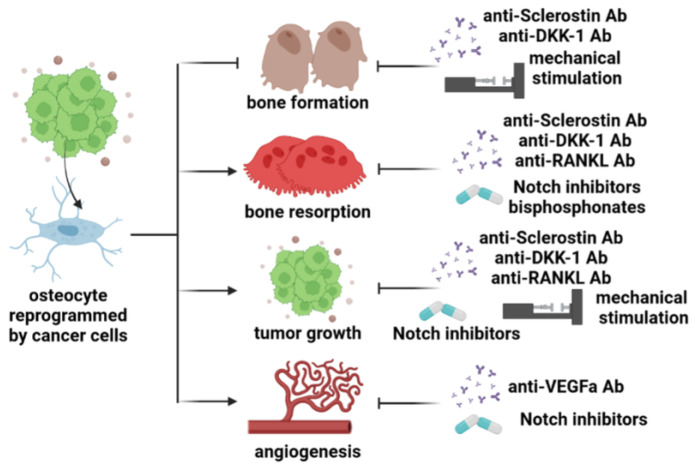
Osteocytes and their derived factors are new therapeutic targets to treat bone cancers. Osteocytes sense and respond to mechanical signals and are a major source of Wnt antagonists in bone. Pharmacologic inhibition of sclerostin or DKK-1, or mechanical stimulation, is efficacious in promoting new bone formation and increasing bone mass in hematological and metastatic cancer models. Osteocytes can also contribute to the beneficial skeletal effects of the anti-resorptive effects of anti-RANKL and bisphosphonates in cancer of the bone. Additionally, targeting sclerostin, DKK-1, or Notch signaling also reduces osteoclast formation in some models of one cancer. Neutralizing osteocyte-derived Notch or VEFGA signals could also reduce angiogenesis and tumor growth.

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
