# Peer review of "Osteocytes: New Kids on the Block for Cancer in Bone Therapy"

_cancers, 2023, doi:10.3390/cancers15092645_

Round 1
Reviewer 1 Report
This is a comprehensive review on a timely subject.
It provides a good overview of osteocytes and cancer.
The review is well-balanced and cites the majority of publications in the field.
The figures provide an appropriate summary of key elements.
Author Response
We thank the reviewer for his/her supportive comments.
Reviewer 2 Report
This manuscript is a very nicely written and comprehensive review on the role of osteocytes in cancer. The study of tumor cell-osteocyte cross-talk is a field that has seen huge growth over the past decade, and the authors have focused their review on topics such as osteocyte effects on cancer cell metastasis and growth; osteocyte effects on neo-angiogenesis in tumor microenvironments, and osteocyte effects on osteoclastic and osteoblastic (osteolytic and osteogenic) cancers. The review of the literature is followed by a Future Directions section that explores how current knowledge may be exploited to develop further treatments for bone cancers.
I have no major concerns with the manuscript, but only suggest that for those readers who might not be completely familiar with bone architecture, a few sentences or a paragraph describing osteocyte structure within the lacuno-canalicular system and the pattern of blood vessels in bone would help in visualization of the physical interactions between osteocytes, osteoclasts, osteoblasts, and the bloodstream.
Author Response
We thank the reviewer for the insightful comments on our manuscript. As requested, we have included a paragraph summarizing the lacuno-canalicular system (page 2, lines 92-102)* and vascular system (page 6, lines 262-266)* in bone.
*Page and line numbers refer to the no-markup version of the revised manuscript.
Reviewer 3 Report
The text is not clear, not well organized and many results are reported without any discussion or perspective. There are many shortcuts. For example, the authors report that osteocytes may influence the migration of myeloma cells because it was shown in a mouse model that suppression of osteocytes impaired the migration and homing of myeloma cells from one tibia to the bone marrow of the contralateral limb. But this effect could be mainly indirect through modifications of the bone metabolism and remodeling. The possible role of osteocytes in the development of primary bone tumors should not be confused with the possibility that some osteosarcoma could originate from osteocyte transformation. This part is embedded in considerations about osteocyte-derived factors that could control cancer cell migration.
The data presented in paragraph 2 are so weak that the 3th one begin by: Opposite to early stages of cancer colonization, osteocytes appear to have an active role in metastatic and hematological cancers.
P6 again assertion about mechanical signals is embedded in targeting NOTCH signaling.
Is it really true that osteocytes are the main source of RANKL in adult bone. What about osteoblast precursors, lymphocytes …
Considerations about targeting osteocytes-derived factors (antiRANKL, anti SOST…) should be in a separated paragraph. It is different from role of osteocytes in tumor growth .
In paragraph 4 the relation between hypoxia- and myeloma-induced expression of VEGFA is not clear. Osteocytes live in a hypoxic environment.
As a general question: is it really possible not to consider that osteocytes have similar roles regardless of the type of cancer?
Some errors have to be corrected
Author Response
The text is not clear, not well organized and many results are reported without any discussion or perspective.
R: We thank the reviewer for this comment and apologize for the lack of clarity. We have reorganized the content into subsections and expanded the text to include further discussion of the results presented.
There are many shortcuts. For example, the authors report that osteocytes may influence the migration of myeloma cells because it was shown in a mouse model that suppression of osteocytes impaired the migration and homing of myeloma cells from one tibia to the bone marrow of the contralateral limb. But this effect could be mainly indirect through modifications of the bone metabolism and remodeling.
R: we have expanded this paragraph to discuss further these results (page 4, lines 176-179)*.
*Page and line numbers refer to the no-markup version of the revised manuscript.
The possible role of osteocytes in the development of primary bone tumors should not be confused with the possibility that some osteosarcoma could originate from osteocyte transformation. This part is embedded in considerations about osteocyte-derived factors that could control cancer cell migration.
R: we thank the reviewer for this suggestion. We have moved this paragraph to the Future Directions section (page 12, lines 571-578).
The data presented in paragraph 2 are so weak that the 3th one begin by: Opposite to early stages of cancer colonization, osteocytes appear to have an active role in metastatic and hematological cancers.
R: We have removed the first paragraph of this section. We also edited the Future Directions sections to indicate the need to investigate the role of osteocytes in bone homing, dormancy, or survival (page 12, lines 563-570).
P6 again assertion about mechanical signals is embedded in targeting NOTCH signaling.
R: We have reorganized the text to separate these two topics into two categories in each section.
Is it really true that osteocytes are the main source of RANKL in adult bone. What about osteoblast precursors, lymphocytes …
R: Using a mouse strain in which RANKL can be conditionally deleted and a series of Cre-deleter strains (Prx1-Cre, Osx1-Cre, Ocn-Cre, Dmp1-Cre, Sost, CD-19, Lck-Cre), several groups demonstrated that osteocytes are an essential source of the RANKL that controls adult bone remodeling under physiological conditions.(1-3) Other cells, including cancer cells, can contribute to the skeletal consequences under pathological conditions. We have clarified this in the text (page 8, lines 377-383).
Considerations about targeting osteocytes-derived factors (antiRANKL, anti SOST…) should be in a separated paragraph. It is different from role of osteocytes in tumor growth.
R: We have reorganized the text to include the discussion of the effects of anti-RANKL and anti-SOST in sections 5 (page 8, lines 385-387) and 6 (page 10, lines 481-506), respectively.
In paragraph 4 the relation between hypoxia- and myeloma-induced expression of VEGFA is not clear. Osteocytes live in a hypoxic environment.
R: we have edited the text to clarify this point (page 6, lines 277-278).
As a general question: is it really possible not to consider that osteocytes have similar roles regardless of the type of cancer?
R: Although osteocytes appear to influence tumor growth, bone resorption, and bone formation in most of the cancers that grow in bone, regardless of cancer type, their impact (positive or negative) on each process varies depending on the cancer type (i.e., Sclerostin actions in primary cancers versus hematological cancers). We have expanded the conclusions section to clarify this point (page 11, lines 519-523).
References
- Xiong J, Piemontese M, Onal M, Campbell J, Goellner JJ, Dusevich V, et al. Osteocytes, not Osteoblasts or Lining Cells, are the Main Source of the RANKL Required for Osteoclast Formation in Remodeling Bone. PLoS ONE. 2015 2015;10(9):e0138189.
- Xiong J, Onal M, Jilka RL, Weinstein RS, Manolagas SC, O'Brien CA. Matrix-embedded cells control osteoclast formation. Nat Med. 10/2011 2011;17(10):1235-41.
- Nakashima T, Hayashi M, Fukunaga T, Kurata K, Oh-hora M, Feng JQ, et al. Evidence for osteocyte regulation of bone homeostasis through RANKL expression. Nat Med. 10/2011 2011;17(10):1231-4.
Round 2
Reviewer 3 Report
the manuscript was improved